# Variation of the Occurrence of Physical Restraint Use in the Long-Term Care: A Scoping Review

**DOI:** 10.3390/ijerph182211918

**Published:** 2021-11-13

**Authors:** Elisa Ambrosi, Martina Debiasi, Jessica Longhini, Lorenzo Giori, Luisa Saiani, Elisabetta Mezzalira, Federica Canzan

**Affiliations:** 1Department of Diagnostics and Public Health, University of Verona, Strada Le Grazie 8, 37134 Verona, Italy; luisa.saiani@univr.it (L.S.); elisabetta.mezzalira@univr.it (E.M.); federica.canzan@univr.it (F.C.); 2Centre of Higher Education for Health Sciences, Azienda Provinciale per i Servizi Sanitari, Via Briamasco 2, 38121 Trento, Italy; martina.debiasi@univr.it; 3Department of Medical Sciences, University of Udine, Viale Ungheria 20, 33010 Udine, Italy; jessica.longhini@uniud.it; 4Santa Maria Del Carmine Hospital, Azienda Provinciale per i Servizi Sanitari, Corso Verona, 4, 38068 Rovereto, Italy; lorenzo.giori@apss.tn.it

**Keywords:** long-term care, physical restraints, prevalence, incidence, scoping review

## Abstract

Physical restraints in the long-term care setting are still commonly used in several countries with a prevalence ranging from 6% to 85%. Trying to have a broad and extensive overlook on the physical restraints use in long-term care is important to design interventions to prevent and/or reduce their use. Therefore, the aim of this scoping review was to analyze the range of occurrence of physical restraint in nursing homes, long-term care facilities, and psychogeriatric units. Pubmed, CINAHL, Ovid PsycINFO- databases were searched for studies with concepts about physical restraint use in the European long-term care setting published between 2009 and 2019, along with a hand search of the bibliographies of the included studies. Data on study design, data sources, clinical setting and sample characteristics were extracted. A total of 24 studies were included. The median occurrence of physical restraint in the European long-term care setting was still high (26.5%; IQR 16.5% to 38.5%) with a significant variability across the studies. The heterogeneity of data varied according to study design, data sources, clinical setting, physical restraint’s definition, and patient characteristics, such as ADLs dependence, presence of dementia and psychoactive drugs prescription.

## 1. Introduction

Physical restraint (PR) is a complex practice, which involves the different health professionals involved in the daily routine of nursing homes; its most shared definition is “any device, material or equipment attached to or near a person’s body and which cannot be controlled or easily removed by the person, and which deliberately prevents or is deliberately intended to prevent a person’s free body movement to a position of choice and/or a person’s normal access to their body” [1,2].

This definition includes a wide range of devices that may be used to contain patients, such as: bilateral bed rails, limb or trunk belts, fixed tables on a chair or chairs that prevent patients from getting up, containment sheets and pajamas [2]; the most frequently reported reasons for their application are essentially three: management of patient agitation [3,4], prevention of falls [5] and for medical devices safety [3,4,6,7]

It is essential to underline that the use of PR does not come without risks: adverse effects can be observed, ranging from bodily injuries, mortality, decreased mobility, to reduced physiological well-being. Nevertheless, it is not associated with a reduction of falls, even if PR use is still justified in clinical practice as a prevention fall strategy [8,9]

Despite this, PR is still commonly used in in long-term care settings of several countries, with a range of prevalence from 6% to 85%. The estimated prevalence of PR use may vary considerably due to several reasons such as: (1) different definitions of physical restraint, (2) different ways to collect and report the data, (3) different characteristics of the care settings (4) clinical variability of patients and (5) different laws and policies [9,10]. The reported prevalence may then further vary depending on the type of restraint used, the estimated time of permanence, and the contextual administration of drugs [9,11,12,13]. Therefore, there is a significant potential for discrepancy in the occurrence of physical restraint use in long-term care that may not always be considered in studies and these relevant differences constitute a challenge for researchers and clinicians, making more complex to identify which factors affect PR use in long-term care.

Many efforts have been made to reduce PR use in a wide range of countries, such as in the United States where a well-designed reform regarding nursing homes (Nursing Home Reform Act OBRA ‘87) [14] has led to a drastic reduction of restraint use. However, the European Union does not currently have a shared law or policy that may be used as a guidance for clinical practice on physical restraints. Thus, as highlighted by Lee et al. (2021) [15], it is essential to understand how culture and economic aspects may have an impact on physical restraint policies of any country.

As this premise suggests, trying to have a broad and extensive overlook on the physical restraint use in long-term care is an important step to accurately measuring and subsequently designing interventions to prevent and/or reduce their use. As Munn and colleagues (2018) [16] stated in their manuscript, performing a scoping review is useful and effective to have an extensive overview of a body of literature. Such a broad and extensive overlook regarding the use of physical restraints in European long term care settings constitutes an essential step to accurately measure and subsequently design interventions to prevent and/or reduce PR use.

Therefore, the aim of this scoping review was to examine the variation of the occurrence (prevalence/incidence) of physical restraint use in the European long-term care settings (nursing homes, long term care facilities, and psychogeriatric units).

## 2. Materials and Methods

This scoping review was based on the design by Arksey and O’Malley [17] with enhancements from Levac et al. [18]. A scoping review is a knowledge synthesis approach focusing on an exploratory research question. It is aimed at mapping key concepts, types of evidence, and gaps in research related to a defined area or field [19]. The method included the following steps: specifying the research question; identifying relevant studies; extracting relevant information; collating and summarizing the information.

The present scoping review was guided by the following research questions: (1) What is the occurrence of the use of physical restraint in the European long-term care setting? (2) What factors contribute to the heterogeneity of data reported in the literature?

### 2.1. Search Strategy

The search strategy included both free text and controlled vocabulary for the concepts of “physical restraint” and “long-term care” (nursing homes, long-term care facilities and psychogeriatric units). The original search was conducted in April 2019. The databases searched were Ovid Medline, PubMed, CINAHL, Ovid PsycINFO. A hand search was conducted of the bibliographies of the included studies. The databases were searched using the following terms: ((nursing home* OR residential care facilit* OR long-term care) and physical restraint*) in combination. 

### 2.2. Identification of Relevant Studies

Studies that defined or assessed physical restraint use in residents living in nursing homes, long-term facilities and psychogeriatric units as an aim, outcome, or covariate, were included. Studies conducted in European countries and published between 1 January 2009 and 30 April 2019, were selected. The motivation for including only the studies published in the last decade was to obtain an updated picture of the phenomenon, to provide guidance to design and act on specific aspects of physical restraint use in the clinical contexts.

Qualitative studies, systematic reviews, meta-syntheses, case series, case reports, conference abstracts, letters, comments, and opinion pieces were excluded; additional exclusion criteria included studies on pediatric population, conducted in hospital setting, geriatric and medical department, intensive care units, home care and residential care facilities, and not written in English, Italian, or Spanish. Studies in which physical restraint prevalence was not reported or could not be computed were also excluded.

Consistently with the scoping review methodology, the quality of included studies was not assessed [17].

Title/abstract and full text screening was performed by two reviewers independently. Disagreements were solved through discussion between the two reviewers or among the full research group.

### 2.3. Data Extraction

Information was extracted regarding the study characteristics including design, year of publication, origin/country of origin (where the study was published or conducted), study population (gender, clinical setting, average age of resident/patients, disease conditions) and sample size (if applicable), data collection, definition, and occurrence of physical restraint. PR occurrence was computed when it was not reported but all the necessary information was available. For this calculation, the definition provided by Bleijlevens and colleagues (“any device, material or equipment attached to or near a person’s body and which cannot be controlled or easily removed by the person, and which deliberately prevents or is deliberately intended to prevent a person’s free body movement to a position of choice and/or a person’s normal access to their body”) [1] was used when a study did not report a definition. Data extraction was completed by one reviewer and checked by a secondary reviewer for accuracy.

### 2.4. Statistical Analysis

The normal distribution of prevalence occurrence was checked using the quantile–quantile plot, Shapiro-Wilk test, considering as acceptable a value lower than |0.5| of Skewness and Kurtosis. The prevalence occurrence was not distributed normally, thus median, and interquartile range have been used as measures to report data. The Mann-Whitney test and the Kruskal-Wallis test have been used when appropriate as non-parametric statistical methods in sub-groups analysis, at a statistical significance level of <0.05, to detect possible associations between prevalence occurrence and categorical variables extracted by study included.

The information from the studies was extracted and imported into a Microsoft Excel 2010 database. Data analysis was conducted using R-statistical software (R Core Team, 2021).

## 3. Results

### 3.1. Summary of Included Studies

Six hundred and nineteen titles were identified and screened, of whom 40 qualified for review and data extraction. Of the 40 studies, 10 that did not report overall occurrence of physical restraint use and six that were study protocols, were excluded. The final sample was constituted by 24 [11,13,20,21,22,23,24,25,26,27,28,29,30,31,32,33,34,35,36,37,38,39,40] studies (Figure 1).

The distribution of the characteristics of the 24 studies that estimated overall prevalence is synthesized in Table 1. The studies were mostly conducted in the Netherlands (33.3%, *n* = 8), published between 2009 and 2013 (58.3%, *n* = 14), cross-sectional or RCT by design (58%, *n* = 14), and predominantly used observation to collect data (38%, *n* = 9).

Moreover, most studies enrolled more than 1000 patients (46%, *n* = 11), predominantly women (100%, *n* = 24), with an average age younger than 85 years (66.7%, *n* = 16) and were conducted in nursing homes and long-term care (75% *n* = 18).

The frequency of physical restraint use was measured both as prevalence (37.5%, *n* = 9) and as incidence (62.5%, *n* = 15); these two measures, for this analysis, were treated the same way.

About 50% of the studies (*n* = 11) considered bed rails as a type of physical restraint. Dementia or cognitive impairment (from mild or severe) and the use of psychotropic drugs (antipsychotics/neuroleptics, benzodiazepines) were both assessed in the 66.6% (*n* = 16) of the studies. In the 45.8% (*n* = 11) of the studies, dependence in activities of daily living (ADLs), was also measured.

### 3.2. Occurrence of Physical Restraint Use

As shown in Figure 2, occurrence of physical restraint use in long-term care was highly variable, ranging from 7.7% to 60.5%. The median occurrence was 26.5% (IQR 16.5–38.5%).

Table 1 shows the median, first and third quartiles of the occurrence of physical restraint use in relation to different characteristics of the included studies.

Concerning European continent, the study conducted in Finland had the highest median occurrence of 52.0% (IQR 52.0–52.0%), while studies (*n* = 2) conducted in Spain had the lowest median occurrence of 14.8% (IQR 13.8–15.8%). In the last decade, a decrease in the physical restraint use has been occurred (27.2%; IQR 17.6–35.7% in the 2009–2013 period vs. 23.2%; IQR 16.6–43.4% in the 2014–2019 period). Nevertheless, no statistically significant difference was found (*p* = 0.841) (Table 1).

With respect to the study design, the quasi-experimental study had the highest median occurrence of 45% (IQR 29.5–59.2%). Among the other study designs, the cross-sectional studies had the highest median occurrence of 29.9% (IQR 23.8–40.7%), while the cohort studies had the lowest median occurrence of 16.8% (IQR 14.5–19.1%), *p* = 0.125.

Studies using the chart review and electronic health records (36.6%; IQR 20.1–43%) had higher median occurrence than studies using other data sources, with standardized scale and interview-based studies reporting the lowest occurrence of 16.8% (IQR 15.2–17.5%), *p* = 0.149.

Studies enrolling a small sample size (33.05%; IQR 16.0–47.9% in the 100–299 patients group) had higher median occurrence than studies with a large sample size (26.3%; IQR 15.1–30.5% in the 1000+ group), *p* = 0.473.

The median occurrence of physical restraint use tended to decrease when the average age of the sample was higher, going from 27.0% (IQR 16.2–46.7%) of the <85 years class, to 23.2% (IQR 18.4–33.9%) of the >85 years class; nevertheless, no statistically significant differences were detected (*p* = 0.742).

Studied conducted in psychogeriatric units (50.8%; IQR 33.4–58.5%) had a statistically significant higher median occurrence than studies conducted in nursing homes and long-term care (22.5%; IQR 16.6–31.8%), *p* = 0.04.

Studies that considered bed rails as a physical restraint had a statistically significant higher median occurrence 36.6%; IQR 28.8–54.3%) than studies excluding them (16.9%; IQR 13.7–23.8%), *p* = 0.002.

There has been a steady increase in the physical restraint use with the worsening of ADLs dependence (7.15%, 39.7%, 53.8%, *p* = 0.002).

Regarding medications, the highest median occurrence of physical restraint use was observed in patients prescribed with psychotropic drugs (not specified) (68.2%; IQR 53.5–70.2%, *p* = 0.014). Studies that defined drug classification reported a median occurrence of physical restraints of 44.7% (IQR 36.0–67.5%) and of 29.1% (IQR 20.8–33.9%) in patients prescribed with antipsychotics/neuroleptics and benzodiazepines, respectively.

## 4. Discussion

Using the scoping review approach, the present study found the occurrence of physical restraint use in long-term care within the European context varying by study setting, methodology, definition of physical restraint, or clinical conditions (e.g., level of ADLs dependence, presence of dementia/cognitive impairment, use of psychotropic drugs). We found studies done in Northern Europe (e.g., Finland and Belgium) or conducted in psychogeriatric units to have a median occurrence higher than 45% while, on the other hand, studies conducted in Southern Europe (e.g., Spain and Italy) or in nursing homes, had a median occurrence of less than 25%.

Study methodology, including design and data sources, had a great impact on the reported occurrence of physical restraint. It is relevant to highlight that most of the included studies in this review had a cross-sectional design. Results from a cross- sectional study may be less able to deepen the phenomenon described as the measurement is conducted only once, compared to longitudinal studies where a series of measurements are performed over a period of time. On the other hand, the opportunity to have six RCTs allowed us to analyze incidence data. Future research on this topic should include prospective longitudinal studies aiming to obtain an accurate occurrence analysis.

The use of the electronic records allows for retrospective studies in healthcare and in our results, the highest median occurrence was observed in the studies which used electronic records, while the lowest median prevalence was found in the exploratory longitudinal studies exploiting direct observation of the patients. It is undeniable that the measurement approach has an impact on the results, and this is confirmed even by the work of Lee and colleagues [15], which suggested to use a direct observation approach or a combination of multiple measurement approaches, in order to not underestimate the occurrence of physical restraint use.

Our review observed a decrease in the occurrence of physical restraint use over the last decade, which may be explained by the publication and dissemination of the guidelines “Promoting Safety: Alternative Approaches to the Use of Restraint”, developed by the Registered Nursing Association of Ontario [41]. Indeed, these guidelines recommend a preventive approach to promote patient safety and, at the same time, guarantee freedom. It is therefore necessary to focus on the assessment of at-risk patients and to implement alternative strategies to the use of physical restraint. In order to implement these changes in clinical practice, it is essential to analyze every professional’s own working context and to design basic, advanced and continuing education interventions relevant to the learning needs of the recipients. However, education represents a necessary, but not sufficient, element to create a restraint-free environment. In this sense, it becomes crucial to implement strategies that promote an organizational culture supporting the implementation of evidence-based guidelines, the active involvement of patients and their caregivers in the care process, the consultation of expert nurses, the availability of alternative strategies, such as low beds or electronic alarm systems.

We found that, compared to younger patients, older patients were more often put on physical restraints, although no statistically significant difference was detected between the three age groups.

The health care setting has a great impact on the occurrence of physical restraints as demonstrated in our study. The psychogeriatric units had the highest magnitude of physical restraint use; this result may be explained by the different clinical characteristics of the patients admitted to this setting, who generally present dementia and behavioral disorders, such as aggressiveness, agitation, disinhibition, and wandering, which are strictly associated to physical restraint use [13,42]. Another possible explanation of the variation of physical restraint use in the different settings may be related to organizational conditions, such as nurse ratio and competence, staff mix and ward characteristics [42]. However, this aspect is very controversial in the international literature; some studies did not find a statistically significant difference between staff intensity and staff mix in relation to the use of restraint [11], other studies found that high workloads and patient’s physical and cognitive disability were moderately associated with the use of physical restraint [43].

The median occurrence of physical restraint use increased in case bed side rails were also considered. Many studies did not consider bed rails as means of restraint although they are extensively used in the clinical practice. Indeed, they are sometimes considered as safety tools, used to reduce the risk of accidentally slipping, rolling, or falling out of bed [43]. In order to improve future research, a shared definition of physical restraints should be provided [40].

Almost all the patients involved in the studies presented dementia and/or cognitive impairment, were prescribed with at least one antipsychotic drug, and had some degree of dependence in ADLs. According to previous studies, being cognitive impaired, highly dependent in ADLs and being prescribed with psychoactive drugs increased the risk of being restrained [43].

It is relevant to highlight that it might be interesting to produce a summary of the PR use occurrence combining the individual effects of each included study. Nevertheless, due to the nature of the scoping review (which is not expected to pool the findings in the form of a meta-analysis) and the extremely high heterogeneity of the included studies, this has not been possible to perform with our study. Future studies need to focus on conducting systematic reviews and specifically to combine the results and give more information in physical restraint use in relation to specific populations and settings, in order to be able to inform policy.

### Strengths and Limitations

The scoping review methodology allows to perform an extensive literature overview according to a rigorous approach. Moreover, our review has included several studies on the phenomenon conducted in a variety of settings, which permitted us to analyze the occurrence of physical restraint in different patients’ subgroups.

However, due to the heterogeneity of the included studies, median and interquartile range has been used to summarize our findings, although they are not the most suitable metrics. The median was preferred over the mean as it is less influenced by the significative range variability founded in the literature reporting the prevalence of physical restraint use. An assessment of methodological limitations or risk of bias of the evidence included in this scoping review was not performed, this could be a limitation. Although, according to the scoping review method, a critically appraisal of the included studies is not expected.

Moreover, some studies published into national medical or health care journals in country languages might have been missed as we did not search in all the available databases. Finally, in this scoping review six studies were unretrieved and this may have affected the occurrence assessment.

## 5. Conclusions

The occurrence of physical restraint in the European long-term care setting is still high (26.4% median). A significant variability emerged across the studies published between 2009 and 2019, with a median occurrence of restraint use ranging from 16.5% (IQR1) to 38.5% (IQR3).

The heterogeneity of data reported in the included studies varied according to study design, data sources, clinical setting, physical restraint’s definition, and patient characteristics, such as ADLs dependence, presence of dementia and psychoactive drugs prescription.

The wide range of methodological and clinical differences make an effective comparison across the involved studies difficult. Further studies need to be focused on the clear description of the factors that are associated with a variability on physical restraint use.

To this aim, particular attention in sample description may be helpful for defining and explaining the different restraint occurrence in the European long-term care.

Moreover, particular attention for future research needs to be focused on the ethical issues concerning physical restraint policy. Indeed, patients or legal representees must be involved in the making decision process and the opportunity of having available data on how many times this happen in real life is essential as it better clarifies who are the health professionals involved in the decision process.

## Figures and Tables

**Figure 1 ijerph-18-11918-f001:**
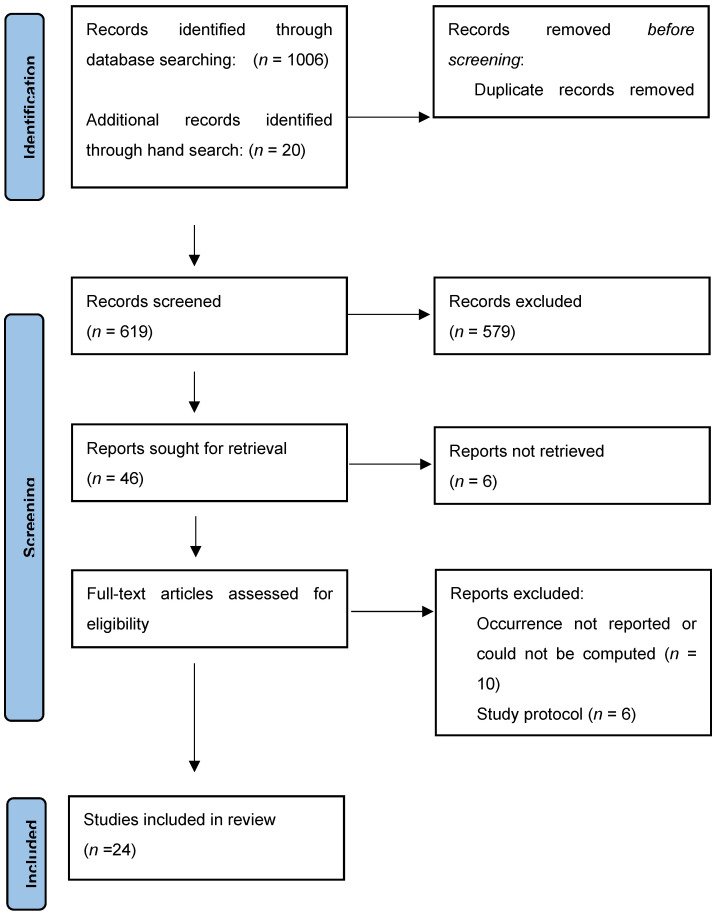
Flow diagram of studies identified, screened, and extracted, PRISMA 2020.

**Figure 2 ijerph-18-11918-f002:**
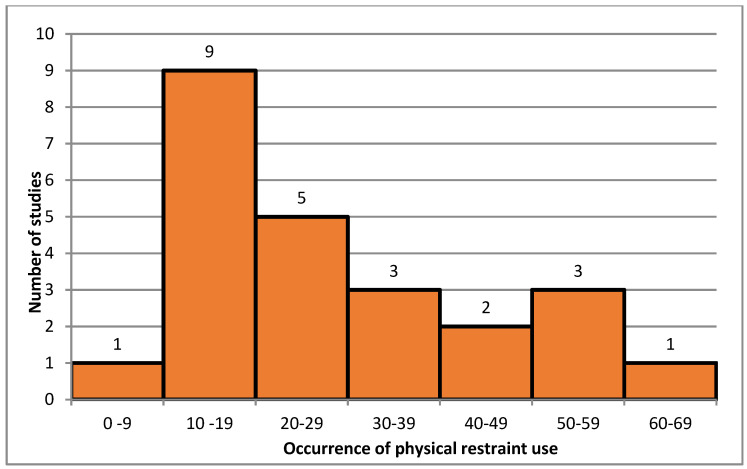
Distribution of overall occurrence of physical restraint use in long-term care.

**Table 1 ijerph-18-11918-t001:** Characteristics of included studies and occurrence of physical restraint use.

Characteristic	N. Studies		Median Occurrence (%)	IQR	*p*-Value
Overall	24	100%	26.39	(6.7–40.4)	
European Countries					-
Netherlands	8	33.3%	41.7	(26.9–57.3)	
Germany	4	16.7%	27.2	(21.6–29.3)	
Norway	2	8.3%	26.5	(21.5–31.6)	
Spain	2	8.3%	14.8	(13.8–15.8)	
Italy	2	8.3%	19.0	(13.7–24.4)	
Belgium	1	4.2%	47.5	(47.5–47.5)	
Sweden	1	4.2%	25.3	(25.3–25.3)	
Swiss	1	4.2%	26.7	(26.7–26.7)	
Finland	1	4.2%	52.0	(52.0–52.0)	
Combinations	2	8.3%	18.3	(17.7–18.9)	
Year of publication					0.841 ^$^
2009–2013	14	58.3%	27.2	(17.6- 35.7)	
2014–2019	10	41.7%	23.2	(16.7–43.4)	
Study design					0.125 ^†^
Cross-sectional	8	33%	29.9	(23.8–40.7)	
Randomized Controlled Trial	6	25%	26,7	(18.7–34.5)	
Quasi-experimental	5	21%	45.0	(29.5–59.2)	
Cohort	5	21%	16.8	(14.5–19.1)	
Data sources					0.149 ^†^
Chart review/Electronic records	7	29%	36.6	(20.1–43.0)	
Observation	9	38%	29.5	(26.0–56.6)	
Standardized scale and interview	4	17%	16.8	(15.2– 17.5)	
Combinations	4	17%	18.2	(16.2–25.9)	
Sample size					0.473 ^†^
100–299	8	33%	33.0	(16.0–47.9)	
300–999	5	21%	25.3	(19.6–47.5)	
1000+	11	46%	26.3	(15.1–30.5)	
Average age of enrolled patients					0.742 ^$^
<85 years	16	66.7%	27.0	(16.2–46.7)	
>85 years	8	33.3%	23.2	(18.4–33.9)	
Sex percentage					-
Mostly females > 50%	24	100%	26.4	(16.7–40.1)	
Mostly males >50%	--	--	-	-	
Health care setting					0.04 ^$^
Nursing home and long-term care	18	75%	22.5	(16.6–31.8)	
Psychogeriatric units	6	25%	50.8	(33.4–58.5)	
Physical restraint role					0.009 ^$^
Outcome	18	75%	31.2	(25.6–46.9)	
Covariate	6	25%	15.7	(13.7–18.5)	
Definition of physical restraint					0.002 ^†^
Bed rails included	11	45.8%	36.6	(28.8–54.3)	
Bed rails excluded	10	41.7%	16.9	(13.7–23.8)	
Not specified	3	12.5%	14.5	(13.9–17.1)	
Presence of dementia in enrolled patients					-
Dementia/cognitive impairment (from mild to severe)	16	66.7%	93.9	(62.1–100.0)	
Not reported	8	33.3%	-	-	
Presence of ADLs dependence in enrolled patients					0.002 ^†^
No	4	16.67%	8.0	(5.7–13.2)	
Mild	9	37.5%	7.2	(5.0–30.0)	
Severe	8	33.3%	39.7	(37.1- 42.5)	
Very severe	11	45.8%	53.8	(21.9- 64.5)	
Not reported	13	54.2%	-	-	
Use of drugs in enrolled patients					0.014 ^†^
Antipsychotics/neuroleptics	7	29.1%	44.7	(36.-67.5)	
Benzodiazepines	7	29.1%	29.1	(20.8–33.9)	
Psychoactive drugs (not specified)	9	37.5%	68.2	(53.5–70.2)	
Not reported	8	33.3%	-	-	

Abbreviations: IQR, interquartile range; ADLs, activities of daily living. ^$^ Mann-Whitney test; ^†^ Kruskal-Wallis test.

## Data Availability

Data are stored at the Department of Diagnostics and Public Health, University of Verona.

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
