# Peer review of "Variation of the Occurrence of Physical Restraint Use in the Long-Term Care: A Scoping Review"

_ijerph, 2021, doi:10.3390/ijerph182211918_

Round 1

Reviewer 1 Report

I would like to thank the authors for the effort they have made in researching this topic. It is a very interesting bibliometric study on published articles of physical restraint. My comments to the authors are as follows:

  1. The quality of statistical reporting and data presentation was poor. I score 3 in a scale from 0 (poor) to 10 (very high). You could improve it. Below are some points that need to be amended.
  2. Please consider further clarification why only studies conducted in Europe were included.
  3. Do not use one sentence paragraphs. You can easily combine paragraphs that focus on the same theme or topic.
  4. Help your readers and clarify why have you not done a systematic review or meta-analysis on this topic? I read this manuscript more as a bibliometric study.
  5. Scopus is an article database indexing specifically European journals, including national journals. I think you should also use Scopus for articles searches.
  6. Page 2, line 91: Consider removing ”also”.
  7. Chapter 2.1.: Please provide the search terms you have used to identify the studies.
  8. Page 3, lines 102-103: I disagree with this view. I think that in bibliometric studies it is important to assess the quality of the articles and the level of evidence. Feel free to comment on this in the discussion section.
  9. Page 3, line 114: Clarify "definition", I think the reference is now not enough for the reader. This is an important point, because it relates to the main response of your study.
  10. Page 3, 2.3. Statistical Analysis: You have only one outcome variable, occurrence of physical restraint (%). Delete references to “each variable”. State clearly that you have calculated median and interquartile values for this outcome variable in sub-groups determined by extracted study characteristics.
  11. Page 3, 2.3. Statistical Analysis: My major hesitation about the statistical analysis is that the Kruskal-Wallis test has been applied to compare groups in a situation where some groups have only one observation. The statistical algorithm does calculate the p-value, but that is not good data analysis. The power of the applied significance test is low. In my opinion, a group with no variation at all (only one article) should not be analysed with significance tests. Please consider combining categories in Table 2.
  12. Page 3, 2.3. Statistical Analysis: Consider replacing “SPSS software” and ”“SPSS for Windows” with “IBM SPSS Statistics software (version 22)”. In addition, you should remove the reference to the previous owner SPSS Inc.
  13. Page 4, lines 140-146: These paragraphs are a repeat from the previous page.
  14. Table 1: What research design is "prevalence"? Also, I think surveys belong to the cross-sectional studies.
  15. Figure 2: Change the categories on the horizontal axis to (0-9, 10-19, 20-29, etc.)
  16. Would ”Occurrence of physical restrain use (%)” serve as a better axis title? In addition, consider replacing “Study count” with “Number of studies”.
  17. Table 2: Please name the statistical test(s) used. Provide the number of articles for each sub-category.
  18. Discussion: The discussion does not read like a careful discussion of the findings. A discussion should also not be a reiteration of the findings, but rather address what the findings mean in the context of the wider field (designing interventions to prevent and reduce their use) and compare them to the published findings of other researchers.
  19. Page 9, lines 249-252: This paragraph fits better in the Methods section. In the limitations paragraph, you could clarify why meta-analytic approach was not used.

Reviewer 2 Report

First of all, congratulations to the author for the work carried out. The study presented is fundamental in the field of long-term care and public health due to the current socio-health conditions and the progressive increase in this type of care. The results are very interesting in terms of proposing measures for improvement in the future.

In the following, I will make a series of considerations:

In Figure 1, I suggest reviewing the figures referring to "Records screened (n=619)" and "Records excluded (n=579)".

On lines 150-153, the text "About 50% of the studies (n=11) considered bed rails as a physical restraint. Dementia or cognitive impairment (from mild or severe) and the use of psychotropic drugs (anti-psychotics/neuroleptics, benzodiazepines, not reported) were both assessed in the 66.6% (n=16) of the studies. In the 54.16% (n=13) of the studies, dependence in activities of daily living (ADLs), was also measured", does not match the corresponding data in table 1.

In table 2: "Randomized Controlled Trial": 26.68 instead of 26,68.

In the Discussion section:

- In line 216 paragraph improve discussion on: perhaps consider that the use of this type of measure requires annotation in the Electronic records, along with the Chart review. It is assumed that it is necessary to note the exact date of placement of the measure and that of its removal. These are measures that require surveillance, periodic control,  etc... so it may be the most reliable measure for recording these situations.

- Another issue to be considered as a future line is the analysis of the existence of informed consent (of the patient if he/she was capable, or of his/her legal representatives), as well as the medical prescription, for the application of this type of measures. This is a very controversial issue since it jeopardizes the fundamental rights of individuals.

Round 2

Reviewer 1 Report

The revised manuscript addresses most of my previous concerns. I am pleased that the authors have appreciated my comments and have amended their manuscripts accordingly. However, I still have some minor comments for the authors.

  1. The statistical reporting and data presentation have improved considerably. However, I suggest that you remove "* p < .05”" from the footnote of Table 1. It is unnecessary since you have reported actual p-values.
  2. Citing Munn et al (2018) now helps readers to assess why scoping review was a valid method for your research problem compared to systematic review. But it does not clarify why you have not done meta-analysis. It is not a limitation that you have not done meta-analysis. Readers could still ask why you not combined the individual effects (median occurrence) of the 24 studies addressing the same question to produce a summary occurrence. Was meta-analysis at all possible? If yes, would it be appropriate? Sometimes meta-analysis is not feasible when there are important differences between the studies in terms of population, study designs, settings, outcomes, quality, or some other factors. Pooling the findings with meta-analytic approach is useful only, if the original studies are similar in characteristics and homogeneous in outcomes. Please highlight this point of view in the discussion.
  3. Scopus does not index grey literature. In addition, European journals that publish scientific articles in their own national languages are not grey literature. Please clarify in the limitations that you may have missed some studies published in national languages in national medical or health care journals.
  4. There are still several one sentence paragraphs that you could easily combine with the following paragraphs.
